Detection of bacterial contaminants and hybrid sequences in the genome of the kelp Saccharina japonica using Taxoblast

http://orcid.org/0000-0001-7987-7523 Dittami Simon M. 1 simon.dittami@gmail.com
http://orcid.org/0000-0001-6354-2278 Corre Erwan 2
1 UMR8227—Sorbonne Universités CNRS UPMC, Station Biologique de Roscoff , Roscoff, Brittany , France
2 FR2424—Sorbonne Universités CNRS UPMC, Station Biologique de Roscoff , Roscoff, Brittany , France
Morrison Hilary
Electronic publication date: 2017 Nov 17
Publication date: 2017
Volume: 5
Electronic Location ID: e4073
Received 2017 Jul 27; Accepted 2017 Oct 30
Copyright: © 2017 Dittami and Corre
Copyright year: 2017
Copyright holder: Dittami and Corre
License: This is an open access article distributed under the terms of the Creative Commons Attribution License, which permits unrestricted use, distribution, reproduction and adaptation in any medium and for any purpose provided that it is properly attributed. For attribution, the original author(s), title, publication source (PeerJ) and either DOI or URL of the article must be cited.
License URL: https://creativecommons.org/licenses/by/4.0/

Keywords: Basic local alignment search tool (BLAST), Horizontal gene transfer, Contaminating sequences, Genome assembly, Brown algae

Funding: French Government via the National Research Agency ANR-10-BTBR-04 This work received support from the French Government via the National Research Agency investment expenditure program IDEALG (ANR-10-BTBR-04). The funders had no role in study design, data collection and analysis, decision to publish, or preparation of the manuscript.

==============================
Modern genome sequencing strategies are highly sensitive to contamination making the detection of foreign DNA sequences an important part of analysis pipelines. Here we use Taxoblast, a simple pipeline with a graphical user interface, for the post-assembly detection of contaminating sequences in the published genome of the kelp Saccharina japonica. Analyses were based on multiple blastn searches with short sequence fragments. They revealed a number of probable bacterial contaminations as well as hybrid scaffolds that contain both bacterial and algal sequences. This or similar types of analysis, in combination with manual curation, may thus constitute a useful complement to standard bioinformatics analyses prior to submission of genomic data to public repositories. Our analysis pipeline is open-source and freely available at http://sdittami.altervista.org/taxoblast and via SourceForge (https://sourceforge.net/projects/taxoblast).

Introduction

Modern genome sequencing strategies rely strongly on the amplification of low quantities of deoxyribonucleic acid (DNA), making them highly sensitive to even small contaminations in the samples. A study by Longo, O’Neill & O’Neill (2011), for example, has shown almost 1/4th of non-primate genomes available in the national center for biotechnology information (NCBI) databases to be contaminated by repeated elements frequently found in human cells. Samples may furthermore be contaminated by airborne bacteria or other eukaryotes, ingested food, or symbionts living within or attached to the target organism.

The detection of contaminants in genome datasets may be accomplished pre-assembly, post-assembly, or using a combination of both approaches. Pre-assembly removal of potential contaminants has the advantage of reducing the complexity of the assembly process by producing smaller and more homogenous data sets. A first step may be filtering according to kmer-coverage or according to per-read guanine-cytosine (GC) contents (Schmieder & Edwards, 2011) or applying more advanced binning techniques based on oligonucleotide composition (Teeling et al., 2004; McHardy et al., 2007). This bears the risk of also removing genomic sequences from the target organism, for example, repeated elements or regions resulting from recent horizontal gene transfers, and of missing contaminants with similar properties as the target genome. A complementary approach is to search for potential contaminants based on sequence similarity. Here the main limitation lies in the quality and completeness of the reference databases. Furthermore, sequence similarity searches may be time-consuming due to the amount of raw data to treat and are classically limited to smaller sets of contaminants such as vectors or individual species (e.g., to screen for human DNA contaminants), or ribosomal sequences. However, the recent development of fast search and classification algorithms that do not rely on marker genes and are dedicated to the analysis of (mainly microbial) metagenomes, such as RITA (MacDonald, Parks & Beiko, 2012), KRAKEN (Wood & Salzberg, 2014), or Kaiju (Menzel, Ng & Krogh, 2016) (see Sangwan, Xia & Gilbert, 2016 for a recent review), could also be applied to standard genome datasets with the aim of identifying contaminations prior to assembly. Just as for kmer- and GC contents-based approaches the limitation remains that foreign sequences recently integrated into the host genome, may be falsely removed.

Post-assembly cleaning of genomic sequences is frequently performed once manual annotation reveals the presence of contaminants (Cock et al., 2010; Olsen et al., 2016), and does not impact non-contaminated scaffolds. As for pre-assembly approaches, criteria to remove sequences are properties such as GC contents and read coverage (Kumar et al., 2013) or sequence similarity (Cock et al., 2010; Collén et al., 2013; Olsen et al., 2016). As mentioned above, GC- and coverage-based approaches will fail to detect contaminants that resemble the target organism with respect to these characteristics, while the performance of sequence-similarity based approaches is strongly dependent on the quality of the reference database. A key advantage of post-assembly approaches, however, is that they allow for the detection of recent horizontal gene transfer events as long as these elements are embedded into scaffolds containing also host DNA.

Here we used a simple analysis pipeline based on multiple similarity searches to detect potential contaminating sequences in the published and assembled genome of the kelp species Saccharina latissima. We show that, in our case, the approach of splitting scaffolds into small fragments prior to blast searches resulted in the identification of several hybrid scaffolds and of approximately 8 Mbp of contaminant bacterial sequences that had previously been missed.

Methods

Two approaches are frequently used for the sequence-similarity based detection of potential contaminants: one is to perform searches with nucleotide sequences (either as blastn against a nucleotide database or blastx against a protein database), but if such searches are carried out with the entire scaffolds, in our experience, they may be biased by highly conserved regions, which frequently have very little discriminatory power (transposons, virus insertions etc.). Alternatively, protein-based searches may be performed with all predicted proteins of a genome against a reference database (frequently the NCBI non-redundant (nr) protein database or uniref90). Based on these results, scaffolds for which most proteins have best hits with expected contaminants are removed. The advantage of the protein-based approach is that, if there are several predicted proteins on a scaffold, then all of them can be analyzed independently, and the results combined to reach a conclusion. Furthermore, protein sequences are more conserved than nucleotide sequences thus making it easier to find hits. The main disadvantage, however, is that this approach depends on the available protein predictions. Not all DNA encodes proteins, and especially bacterial contaminations in eukaryote genomes frequently lack protein predictions. This lack can be partially explained by the heavy reliance of gene calling software on ribonucleic acid sequencing (RNAseq) data, which, in turn usually includes PolyA-selection during library preparation, removing bacterial messenger RNA from the final dataset.

Here we developed and used a pipeline called Taxoblast that constitutes a compromise between both approaches. We worked with nucleotides to eliminate the impact of protein predictions, but in a first step (Fig. 1) each scaffold of the S. latissima genome was split into small fragments of, for example, 500 base pairs (bp). Then, each of these fragments was compared individually to a reference database. This way each sequence fragment had the same weight in the analysis, that is, analyses were not biased towards few highly conserved elements as may have been the case for blast searches with a single long query sequence. Internal tests aiming to identify bacterial sequences in algal genomes have exhibited very little differences between blastx searches against the NCBI protein nr (non-redundant) database and blastn/discontinuous megablast searches against the NCBI nucleotide nt database, except that blastx searches were significantly slower. Regular (continuous) megaBLAST searches are designed specifically for highly similar sequences with >95% sequence identity (McGinnis & Madden, 2004) and were not sufficiently sensitive to detect potential bacterial contaminants in the genome of brown algae; this algorithm, may, however, be preferable when attempting to detect well-known contaminants such as human DNA. In a third step, the taxa corresponding to the best blast hits were summarized for each scaffold to identify potential foreign DNA fragments (note that the taxonomic resolution used in our pipeline can range from specific species to entire domains). As a rule of thumb, we considered that sequences with >90% of hits with potential contaminants were likely to be contaminants, whereas scaffolds with >90% of hits belonging to the target group were most likely not.

Figure 1 Overview of the Taxoblast pipeline (A), the corresponding graphical user interface (B) and the generated output (C).

Results and Discussion

Taxoblast was used to screen the published genome of Saccharina japonica for potential traces of foreign DNA. This genome had already been treated post-assembly to remove bacterial contamination using blastx searches of entire scaffolds against the NCBI nr protein database in combination with analyses of ORF density and GC contents (Ye et al., 2015). Only scaffolds >2kb were considered for our analyses, and blastn searches were carried out against the NCBI nucleotide (nt) database (version July 19th, 2016) with an e-value cutoff of 0.01. Please note that decreasing this cutoff to 1e-5 had virtually no impact on the results obtained.

The total calculation time for the blast analyses on the cluster of the “Analysis Bioinformatics for Marine Science” (ABIMS) platform (http://abims.sb-roscoff.fr) was five days. The other steps of the analysis took less than 10 min on a 2.8 GHz Intel Xenon desktop computer. This reasonable total runtime was achieved although the megablast algorithm used in our pipeline is 909 times slower than Kraken and 83 times slower than MetaPhlAn (Wood & Salzberg, 2014), mainly because our pipeline analyzes only the assembled genome data rather than running on raw reads. Taxonomic summaries were calculated first with the aim of distinguishing eukaryote (taxon 2759) from bacterial (taxon 2) sequences, and in a second round to distinguish diatom (taxon 2836) from brown algal (taxon 2870) sequences. Both bacteria and diatoms are known to form algal biofilms (Lage & Graça, 2016) and are thus common contaminants in cultures of marine algae. Despite previous cleaning efforts undertaken prior to the publication of the S. japonica genome, several bacterial or bacteria-like sequences were found in the published assembly (see Supplemental Information File 1), but we did not identify any potential diatom sequences.

In total, 894 scaffolds >2 kbp (of 6,985) corresponding to 7.96 Mbp of sequence information were identified as bacterial based on the 90% bacterial hits threshold defined above (Fig. 2A). Please note that only 148 of these 894 scaffolds were predicted to comprise protein coding sequences (201 proteins in total), underlining the usefulness of blastn searches rather than blastp or blastx searches, at least in some cases. Nine of the scaffolds classified as bacterial contained 16S fragments and could be assigned to different bacterial taxa using RDP classifier (Wang et al., 2007). These were Haliea (scaffold3770), Methylophaga (scaffold4403), an unclassified Gammaproteobacterium (scaffold6634), Lewinella (scaffold4968), an unclassified Proteobacterium (scaffold4223), Marinoscillum (scaffold3608), Pseudonocardia (scaffold7323), an unclassified Alphaproteobacterium (scaffold5114), and Flexibacter (scaffold4987). We also manually examined the five longest potential bacterial sequences without 16S and confirmed the automatic classification: for scaffold2350 (50 kbp) best hits were found with Rhodobacteraceae; scaffold2282 (54 kbp) was found to be most similar to known sequences of Actinobacteria; scaffold2573 (40 kbp) contained several conserved regions with sequences from Planctomycetes; scaffold2615 (39 kbp) was clearly part of a Methylophaga genome (also detected by RDP classifier, see above); and scaffold2647 (37 kbp) had best hits with genomes of different Alphaproteobacteria. Finally, we examined the distribution of GC contents for all 894 scaffolds classified as bacterial and compared it to that of the remaining algal and unclassified scaffolds using the prinseq server version 0.20.4 (Schmieder & Edwards, 2011) (Fig. 2B). While the algal reads exhibited a unimodal distribution with a narrow peak in GC contents at ca. 49%, reads classified as bacterial had a wide multimodal distribution with peaks at 30, 38, 53, and 64% GC. This supports the hypothesis of diverse phylogenetic origins of the scaffolds classified as bacterial (Kumar et al., 2013).

Figure 2 Taxoblast analysis of the S. japonica genome.

Application of the Taxoblast pipeline to identify potential bacterial sequences in the published S. japonica genome (Ye et al., 2015). (A) shows the percentage of bacterial/eukaryote blast hits over the 6,731 scaffolds >2 kbp with blast hits (254 scaffolds >2kbp had no hits). Dotted lines show the 90% cutoff proposed to consider a sequence as “contaminant”. (B) and (C) illustrate the different distribution of GC contents in the sequences considered bacterial, and those considered eukaryotic or unclassified.

Taxoblast also highlighted 1,060 scaffolds (corresponding to a total of 82 Mbp of sequence information) that could not be attributed to either bacteria or eukaryotes with ≥90% of the best blast hits. These scaffolds may comprise assembly artifacts, for instance contaminating sequences that have been assembled with sequences of the target species, as well as scaffolds with insertions resulting from recent horizontal gene transfers. For the purpose of illustration we have selected the two scaffolds with the highest numbers of blast hits from this category, and manually examined them.

The first potential “hybrid” scaffold, scaffold159, is approximately 500 kbp long. 557 blast hits were obtained for the individual 500 bp-segments of this scaffold, 26% of which (towards the 5’ end of the sequence) were with bacteria. Consequently, we uploaded scaffold159 to the GC-Profile server (Gao & Zhang, 2006), which searches for variations in the GC contents within a sequence, revealing one segmentation point in the scaffold: bases 1–79,743 exhibited an average GC content of 40.7% while for bases 79,744–504,453 the average GC content was 49.9%. Manual examination of both parts of the scaffold separately confirmed that the first part was highly similar to a cyanobacterial genome of the genus Stanieria (79% sequence identity in aligned regions), whereas the second segment aligned with genomic sequences of the brown alga Ectocarpus siliculosus. For the second examined scaffold, scaffold248 with a total length of 422 kbp, the situation was similar, except that the segmentation point was found at position 151,859 and that the first part of the sequence was highly similar to published genomes of bacteria belonging to the Flammeovirgaceae. The corresponding GC contents were 51.6% for the first (bacterial) part of the sequence and 50.3% for the rest.

Interestingly, neither of the two examined scaffolds contained undefined bases (Ns) between the bacteria-like parts and those conserved with other brown algae, but more information would be required to distinguish between recent horizontal gene transfers and assembly errors. In particular, differences in sequencing coverage between different parts of the scaffold could provide an indication for the latter, but unfortunately, no genome browser is available for S. latissima yet. In the case of similar coverage, ultimately polymerase chain reactions would be required to confirm horizontal transfers. Although not all of the results obtained via our pipeline are as clear as these examples, the presented data demonstrate that the current version of the S. japonica genome still contains substantial amounts of bacterial contamination. We therefore suggest removing at least the 864 scaffolds with ≥90% bacterial hits from the genome prior to further analyses.

Conclusion

We used a simple analysis pipeline based on multiple blastn searches with small sequence fragments to detect sequences of different phylogenetic origins in the published genome of the kelp S. latissima. This procedure highlighted several contaminating bacterial sequences as well as hybrid scaffolds that should not or only partially be considered in future analyses of this genome. To facilitate similar analyses in other organisms, we have added a graphical user interface to our pipeline and made it publicly available at http://sdittami.altervista.org/taxoblast and via SourceForge. The output format of this pipeline is a tab-separated text file compatible with most spreadsheet programs. This makes it easy to combine results with other sources of information (e.g., GC content or coverage) or integrated alignment-free tools such as the recently published PhylOligo (Mallet et al., 2017). Sequences requiring further attention such as the aforementioned hybrid scaffolds can be easily identified and further investigated either manually or using semi-automatic pipelines such as PhyloGena (Hanekamp et al., 2007). Currently an important limitation of our approach is that for larger (i.e., eukaryote) genomes, blast searches still need to be run on a dedicated blast server. One possibility will be to replace BLAST by recent and accelerated algorithms such as PLAST (Nguyen & Lavenier, 2009), HS-BLASTN (Chen et al., 2015), or NSimScan (Novichkov et al., 2016) that are able to output the standard tabular BLAST format. Although the usefulness of our pipeline in other systems, for example, for more closely related contaminant genomes remains to be tested, the presented case of the S. japonica genome underlines the importance of systematically including post-assembly pipelines for the detection of contaminant sequences in genome projects.

Supplemental Information

Supplemental Information 1 Supplemental Information File 1.

Taxoblast output when run with the S. japonica genome (scaffolds >2 kbp) to distinguish between eukaryote and bacterial sequences.

Click here for additional data file.

We would like to thank the ABIMS platform (http://abims.sb-roscoff.fr) for access to their computing facilities and C. Boyen for critical reading of the manuscript.

Additional Information and Declarations

Competing Interests

Author Contributions

Data Availability

The authors declare that they have no competing interests.

Simon M. Dittami analyzed the data, wrote the paper, prepared figures and/or tables.

Erwan Corre contributed reagents/materials/analysis tools, reviewed drafts of the paper.

The following information was supplied regarding data availability:

TaxoBlast (including the source code) is freely available at http://sdittami.altervista.org/taxoblast and via SourceForge (https://sourceforge.net/projects/taxoblast/).

The S. japonica genome analyzed is available in Genbank under accession number JXRI00000000.1 (Ye et al., 2015).

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
