# Peer review of "Detection of bacterial contaminants and hybrid sequences in the genome of the kelp Saccharina japonica using Taxoblast"

_PeerJ, doi:10.7717/peerj.4073_

## Round 0.1 · original submission · Major Revisions

One of the three reviewers recommended rejection. However, if you are willing to make some significant additions to the manuscript and address the criticisms of all the reviewers, the article may be publishable.

Most significantly, you will need to provide a comparison of your methods with at least one other informatics tool that performs a similar function. Kraken is one choice. You also need to analyze additional datasets.

Although two of the reviewers thought that the quality of the writing was satisfactory, I tend to give more weight to reviewer 1's opinions in this area. Reviewer 1 is a native English speaker and appears to have read the manuscript quite carefully.

I hope that you will be willing to make these revisions.

Reviewer 1 ·

Basic reporting

Unclear, ambiguous writing and need some language editing

Scientific writing need significant effort. e.g. no references, poor hypothesis and synthesis

Experimental design

No sufficient evidence/detail to highlight that this tool is anywhere near the state of the art in this category

Validity of the findings

Zero Novelty

Data is not robust, e.g. used only one genome, no benchmarking

Additional comments

Thanks for your efforts. According to me this paper/tool is not publication ready in its current form and needs a significant effort (especially in benchmarking and scientific writing)

Major

Where is the benchmarking? You should compare the tool with the existing mainstream tools and provide some comp. Stats.

Why using only one genome? Why not Bacterial Genomes?

Since authors have mentioned that this tool can be used for “other genomes” let me give some bacterial genome scenarios. (a) If I have multiple genomes from diff. strains of single species, do you think this tool can distinguish the intra-species contamination? Here the strain means whole genome based ANI >99%? I don't think its possible. (b) How can we use this tool to identify the contamination in genomes assembled across shotgun microbiome denovo assemblies? where we have very few ref. Genomes?


You have used word “matched” all across the text, mentioning the blast hits. Honestly the word “matched” is very unclear to me, what is the cutoff that makes you decide that its a match?

Overall, its not a well written paper and need some significant effort.

Line 79 “but if such searches are carried out with the
entire scaffolds, they may be biased by highly conserved regions, which frequently have very little
discriminatory power (transposons, virus insertions etc.)”

This statement need ref.

Line 81 “ Alternatively, protein-based searches may be
performed with all predicted proteins of a genome against a reference database (frequently NCBI nr or
uniref90).”

where is the ref.? Who created these two categories?

Line 83 “Based on these results,”

what results? Please re-frame

Line 84 “The advantage of this approach”

what approach? Please write the text clearly.

Line 141 “also detected via a 16S sequence in a different scaffold, see above”

According to me the above line makes no sense, what is “via a 16S sequence” do you mean a blast hit? If yes, what is the aignment parameters? e.g. length, e-value?

Line 142 “large regions that were homologous”

what is the defination of large here?

Line 147 “This clearly supports the hypothesis of diverse phylogenetic origins of
the scaffolds classified as bacterial.”

The above statement needs ref.

Line 150 “the 90% threshold”

what is that threshold

Line 151 “These scaffolds
may comprise assembly artifacts the have merged contaminating sequences with sequences of the
target species as well as recent horizontal gene transfers.”

The above sentence is very confusing, poorly written and has no evidence mentioned in this paper.

Line 152 “It is not the aim of this publication to re-
analyze the S. japonica genome, but for the purpose of illustration we have selected the two scaffolds
with the highest numbers of blast hits from this category, and manually examined them.”

If its not the aim why to do this illustration at the first place?

Line 155 “The first, scaffold159, is approximately 500 kbp long and has 557 blast hits for different 500 bp-
segments, 26% of which (towards the beginning of the sequence) are with bacteria.”

above senetence needs formatting and reframing

Line 157 “GC profile server”

I think its GC-Profile server

Reviewer 2 ·

Basic reporting

The manuscript and the provided software-tool Taxoblast addresses the need of molecular biology scientists to analyse scaffolds from genome assemblies for possible contamination or lateral gene transfer. It describes the tools functioning and features and explains the utility based on an example assembly from the brown alga Saccharina japonica. Advantages of the method in relation to screening reads before assembly are shown. Overall the writing is good instructure, language and well understandable in expression.
The provided tool is described as helpful for biologists that do not have capacity to perform similar analyses as effectively as with the tool (abstact: "usable also by researchers with little background in informatics"). This is an important added value of bioinformatics tools in general, thus not worth mentioning for publishing of this specific tool.

Experimental design

Validating the tool by an example data set was done carefully and has led to new insight, useful for improving the publication of the chosen genome data.

Validity of the findings

The findings are well documented, and the summary given in statistics data is complete.

Additional comments

I recommend the paper for publication after minor revision indicated above.

Reviewer 3 ·

Basic reporting

No comment (see below).

Experimental design

No comment (see below).

Validity of the findings

No comment (see below)

Additional comments

The authors describe in the manuscript a pipeline named “Taxoblast”, which uses a BLAST-based approach and predefined taxa to identify contaminants, as well as hybrid sequences, in assemblies of genomes. The pipeline is publically available. As a case study, the authors examined the genome assembly of the kelp Saccharina japonica and identified some bacterial contaminants as well as bacteria–kelp hybrid contigs.

The manuscript is well-written and easy to follow. The described approach is simple, and it is laudable that the pipeline is freely available.

However, the manuscript reads rather like a case study. It remains uncertain whether the kelp genome harbours particular problems, or whether similar problems occur in other datasets. For example, the authors state that internal tests found very little differences between blastn and blastx searches. However, this may not apply for other genomes, in which the possible contaminants are less well-known, or less diverged from the host (for example eukaryote parasites). I expect that in these cases, a blastx-based approach has clear advantages.

It would also be useful to compare the present approach with some other programs or platforms, e.g. KRAKEN, which are commonly used for identification of contaminants.

Other issues:

l. 40: I would not call a paper published in 2011 as “recent”. Moreover, it would be useful if the authors cite a study showing that contaminants are an even more eminent issue in NGS-derived genomes.
l. 50 (and other places throughout the manuscript): Something went wrong with the in-text citations, for example here: “(e.g. [3,4])” .
l. 98-99: I think the authors mean: "blastx searches against the National Center for Biotechnology Information (NCBI) _protein_ nr database and blastn searches against the NCBI _nucleotide_ nr database". nr = non-redundant.

---

## Round 0.2 · accepted · Accept

The manuscript has been recast to describe a case study rather than emphasizing a novel method of analysis, which satisfies the major criticisms of the original reviewers.

Reviewer 3 ·

Basic reporting

no comment

Experimental design

See below.

Validity of the findings

no comment

Additional comments

The authors have submitted a revised version of their manuscript that describes the pipeline named “Taxoblast”, a BLAST-based approach to identify contaminants and hybrid sequences in contigs.

The authors have added a brief benchmark estimate and have corrected minor errors. The manuscript has significantly improved, and I acknowledge the rationale behind the study. I would have preferred a broader approach that compares, e.g., the results obtained with different methods and with different data. However, the paper is sound as it is.